# On the Development of a New Flexible Pressure Sensor

**DOI:** 10.3390/mi15070847

**Published:** 2024-06-29

**Authors:** Florian Pistriţu, Marin Gheorghe, Marian Ion, Oana Brincoveanu, Cosmin Romanitan, Mirela Petruta Suchea, Paul Schiopu, Octavian Narcis Ionescu

**Affiliations:** 1National Institute for Research and Development in Microtechnologies-IMT Bucharest, 077190 Bucharest, Romania; florian.pistritu@imt.ro (F.P.); marian.ion@imt.ro (M.I.); oana.brincoveanu24@gmail.com (O.B.); cosmin.romanitan@imt.ro (C.R.); 2Doctoral School of Electronics, Telecommunications and Information Technology, National University of Science and Technology POLITEHNICA Bucharest, 061071 Bucharest, Romania; schiopu.paul@yahoo.com; 3NANOM MEMS SRL, Strada George Cosbuc 9, 505400 Brasov, Romania; maringhe@nanom-mems.com; 4Center of Materials Technology and Photonics, School of Engineering, Hellenic Mediterranean University, 71410 Heraklion, Greece; 5Faculty of Mechanical and Electrical Engineering, Petroleum and Gas University of Ploiesti, 100680 Ploiesti, Romania

**Keywords:** pressure sensors, flexible electronics, elasticity, wearability

## Abstract

The rapid advancement of the Internet of Things (IoT) serves as a significant driving force behind the development of innovative sensors and actuators. This technological progression has created a substantial demand for new flexible pressure sensors, essential for a variety of applications ranging from wearable devices to smart home systems. In response to this growing need, our laboratory has developed a novel flexible pressure sensor, designed to offer an improved performance and adaptability. This study aims to present our newly developed sensor, detailing the comprehensive investigations we conducted to understand how different parameters affect its behaviour. Specifically, we examined the influence of the resistive layer thickness and the elastomeric substrate on the sensor’s performance. The resistive layer, a critical component of the sensor, directly impacts its sensitivity and accuracy. By experimenting with varying thicknesses, we aimed to identify the optimal configuration that maximizes sensor efficiency. Similarly, the elastomeric substrate, which provides the sensor’s flexibility, was scrutinized to determine how its properties affect the sensor’s overall functionality. Our findings highlight the delicate balance required between the resistive layer and the elastomeric substrate to achieve a sensor that is both highly sensitive and durable. This research contributes valuable insights into the design and optimization of flexible pressure sensors, paving the way for more advanced IoT applications.

## 1. Introduction

The need for wearable pressure sensors has significantly increased in recent years due to their extensive application in the medical and Internet of Things (IoT) domains. These advanced sensors are crucial in monitoring and providing valuable data regarding human motion and physiological signals. They play an essential role in capturing human movement dynamics, including intricate muscle activities [1] and specific joint movements such as arm motion [2] and knee bending [2,3]. Additionally, these sensors are essential for monitoring vital signs such as pulse rate [4,5,6,7], respiration [1], and even phonation detection [2,8], which is the ability to detect and analyze sounds produced by the vocal cords. The information obtained from wearable pressure sensors is invaluable in numerous applications. In the medical field, these sensors contribute to more accurate diagnosis, patient monitoring, and the development of personalized treatment plans. For instance, tracking muscle movement and joint activities can aid in the rehabilitation process, allowing for real-time feedback and adjustments to therapy protocols. Monitoring pulse rate and respiration provides critical data for managing cardiovascular and respiratory conditions, ensuring timely interventions. Furthermore, in the realm of the IoT, these sensors enable enhanced human–machine interactions and smart environments, offering a seamless integration of technology into daily life. The manufacturing technology for flexible electronics, which underpins the development of wearable pressure sensors, is generally based on advanced printing techniques. Inkjet printing [9,10,11,12] is widely used due to its precision and ability to create the fine patterns necessary for high-performance sensor components. This method allows for the deposition of functional materials onto flexible substrates, forming the essential layers of the sensors. Screen printing [12,13,14,15], another prevalent technique, is known for its cost-effectiveness and scalability. It involves transferring ink through a mesh screen to create thicker and more durable sensor layers, suitable for various flexible electronic applications. The combination of these advanced manufacturing technologies ensures the production of reliable, flexible, and high-performance wearable pressure sensors. By leveraging inkjet and screen printing methods, manufacturers can achieve the necessary precision, efficiency, and adaptability required to meet the growing demands in medical and IoT applications. As a result, these sensors continue to evolve, providing increasingly sophisticated solutions for monitoring and enhancing human health and interaction with technology.

Flexible substrate-based pressure sensors have been extensively studied. For illustration, Congcong Yang et al. designed a film with a double surface structure to create a sandwich resistive pressure sensor, demonstrating excellent parameters including a high sensitivity (77.78 kPa^−1^, 24 Pa minimum detection), a wide detection range (0.024–230 kPa), a fast response time (30 ms), and a high reliability over 5000 repetitive cycles. Mohammed Mohammed Ali et al. [16] developed a printed strain sensor by screen printing a silver nanowire (Ag NW)/silver (Ag) flake composite on a flexible and stretchable thermoplastic polyurethane (TPU) substrate. The sensor was tested in two configurations, as follows: a straight line and a wavy line. The average resistance changes over 100 cycles were 104.8%, 177.3%, and 238.9%, and over 200 cycles were 46.8%, 141.4%, and 243.6% for elongations of 1 mm, 2 mm, and 3 mm, respectively. This sensor is intended for biomedical and civil infrastructure applications. Daniel Gräbner et al. [17] examined the influence of electrode geometry on the performance and cross-sensitivity to strain in screen-printed pressure sensors. Their findings indicate that pressure sensitivity increases with the number of interdigital electrodes, while temperature cross-sensitivity remains unaffected by electrode configuration. Dinesh Maddipatla et al. [18] developed a carbon nanotube (CNT)-based pressure sensor for pressure monitoring applications. The flexible capacitive pressure sensor was fabricated using screen printing technology, employing CNT ink for the top and bottom electrodes and PDMS for the non-conductive dielectric layer. The sensor exhibited an 8.2% change in capacitance at a maximum detectable pressure of 337 kPa, a 0.021% change in capacitance per kPa, and a correlation coefficient of 0.9971. Potential applications include sports, military, robotics, automotive, and biomedical fields. Sepehr Emamian et al. [19] successfully fabricated a fully printed piezoelectric-based touch sensor device on flexible polyethylene terephthalate (PET) and paper substrates using screen printing. The device utilized silver (Ag) ink and sandwiched a screen-printed polyvinylidene fluoride (PVDF) piezoelectric layer between the printed Ag top and bottom electrodes. Sensitivities of 1.2 V/N and 0.3 V/N, with correlation coefficients of 0.9954 and 0.9859, were achieved for the PET- and paper-based sensors, respectively. These sensors are suitable for both touch- and force-based applications. As shown by these examples, pressure sensors fabricated using this technology feature flexible PDMS substrates [20,21], contrasting with the rigid substrates used in traditional electronics. Screen printing remains the most prevalent method for printed electronics, as highlighted in the review by Saleem Khan et al. [22].

A new pressure sensor developed in our laboratories features a novel architecture, as illustrated in Figure 1. To enhance sensitivity, the substrate consists of two layers of PDMS with micro-pyramids, topped with a sensitive layer printed on Kapton (Goodfellow, Coraopolis, PA, USA).

## 2. Materials and Methods

The micro-structured substrate used in the newly developed flexible pressure sensor was made of PDMS Sylgard 184 (DOW Chemical Company, Midland, MI, USA), to which was added 10% powder of aerogel (Powder aerogel < 0.125 mm–Green Earth Aerogels, Barcelona, Spain). The micro-structured substrate designed was of the micro-pyramidal type, with the pyramids having a base of 1500 µm and a height of 1060 µm. The distance between the pyramids was equal to the length of the pyramid base. To obtain the micro-structured substrate, the moulding technique has been used. The moulds were created via 3D printing using the µMicrofluidics M50 3D printer (CADWORKS3D, Toronto, ON, Canada), together with the Utility 6.0 software. The resin used for mould production was Master Mold for PDMS Devices—3D Printing Resin Photopolymer Resin (Composition: methacrylated oligomer, methacrylated monomer, photoinitiator, and additives, (CADWORKS3D).

The procedure of moulding the micro-structured substrate started with the mixing of the polymer Sylgard 184 with the hardening agent in a ratio of 10 to 1. Then, 10% of powder aerogel was added and the process of mixing continued for 10 min. The mixture was poured on the mould and was introduced into a vacuum oven VO400 (MEMMERT, Schwabach, Germany): 49 L; + 20… + 200 °C; 10…1100 mbar). The hardening treatment was conducted at a temperature of 100 °C for 50 min.

The sensing material deposition was performed by printing with the semi-automatic screen printer device LC-TA-250 Model (LC Printing Machine Factory Limited, Guangzhou-Foshan, China) on Kapton. The screen printing process was conducted by using two types of ink. The first type of screen printing ink (Ink1) is composed of 50% carbon micro-powder (graphite and C black), 10% polymer (binder), and the remaining 40% is a xylene-based solvent. The second version of the screen printing ink (Ink2) is composed of 50% carbon micro-powder (graphite and C black), 10% polymer (binder), 15% TiO_2_, and the remaining 25% is a xylene-based solvent (Chemical materials were sourced from Goodfellow, Coraopolis, PA, USA).

To ensure that the ink components were properly integrated in the printed layers, SEM was used to examine meander resistor prints on a flexible substrate and to analyze their morphology. For this, the Nova NanoSEM 630 device (FEI Company, Hillsboro, OR, USA) was used. The materials used in the inks and the printed resistor were investigated.

Figure 2 below shows some examples of SEM images of nanomaterials used in the ink formulations—Figure 2a–c, as well as the printed layers’ surfaces in Figure 2d,e. As one can see, the printed surfaces are uniform and smooth.

The crystalline structure of nanomaterials used in ink formulations, as well as the printed resistor, were verified using X-ray diffraction analysis (XRD—X-ray Thin-film Diffraction System (XRD)/SmartLab (FN2670N)/ from Rigaku Corporation, Osaka, Japan) and it was observed that their structure is preserved in the printing process.

The pattern of piezoresistive sensors was kept identical for all types of flexible substrates used in printing, varying only the thickness of the ink layer. Thus, we were able to study both the influence of the ink layer thickness and the influence of the flexible substrate on the behaviour of the flexible piezoresistive sensor. After printing the piezo-resistor on the flexible substrate, a measurement of the resistor value was performed to establish the resistance value for each sample prepared. Following the resistance recording, the next step was to connect the terminals to the sensors. The process was carried out by using silver paste LOCTITE ABLESTIK 84-1LMI, (Henkel, Düsseldorf, Germany) and AWG 26 wires (Alpha Wire, Elizabeth, NJ, USA). The polymerization was finalized by applying a heat treatment at 125 °C for 30 min.

### Method of Testing

Figure 3 presented below shows the measurement scheme that was used to test the resistive film assembly with the elastomeric substrate, for compression.

The device used for the compression tests is the Mecmesin MultiTest 2.5I (Mecmesin, Slinfold, West Sussex, UK), together with the Emperor™ Force software v1.18 (Mecmesin, Slinfold, West Sussex, UK). MECMESIN is a single-column computerized tensile testing machine suitable both for metal and non-metallic materials testing. This machine adopts a mechanical and electrical integration design, mainly composed of a force sensor, a transmitter, a microprocessor, a load driving mechanism, and a computer. It has a wide and accurate range of loading speed and force measurement, the ability to measure and control the load, and the displacement has a high precision and sensitivity, but it can also carry out the automatic control test of constant speed loading and constant speed displacement.

Using Mecmesin MultiTest 2.5I, one can test various materials, semi-finished products, and finished products for their tensile strength, compressive strength, and elongation—elongation can be used for peeling, tearing, bending, compression, and other tests; these tests are suitable for metal, plastic, rubber, textiles, synthetic chemicals, wire and cable, leather, and other industries. Due to its capabilities to be programmed in a flexible manner, it could be used easily for cycle testing the pressure sensors. The measurements of the screen-printed resistors were performed with the FLUKE 8846A 6-1/2 Digit multimeter (Fluke, Everett, WA, USA), plus the Pomona 6730–Wide Jaw Kelvin Lead Set test leads (Pomona Electronics, Everett, WA, USA,). The power supply used is Agilent E3648A (Agilent, Santa Clara, CA, USA), a dual output power supply—two 8 V, 5 A or 20 V, 2.5 A. The resistor used in the electrical scheme for testing has a value of 6750 Ω and a tolerance of ±5%.

## 3. Results and Discussion

Several variants (replications) of meander resistor assemblies with flexible substrates have been produced. It was investigated whether the heat treatment at 125 °C for soldering the terminal wires has an influence on the value of the meander resistor or not.

The values obtained before applying the thermal treatment (named in the table column ‘Initial Value’) for the bonding of the terminal wires and the value obtained after the thermal treatment (named in the table column ‘Final Value’) are given in Table 1.

The thermal treatment at 125 °C was found to affect the resistors by reducing the thickness of the resistive layer, which, in turn, increased the resistance values. This may be a consequence of solvent evaporation. Therefore, it is recommended to use a silver paste that requires a significantly lower heat treatment.

The subsequent investigation focused on examining how bending affects the meander resistor, specifically at 45° and 90° angles. The results indicated that ABS (R3) exhibited the highest variation in resistance during bending, followed by PET (R2) and Kapton (R1). Table 2 provides detailed data, where “Material” denotes the substrate material of the meander resistor, “Initial Value” is the resistance value before bending, “Bending 45°” and “Bending 90°” indicate the resistance values at 45° and 90° bends, respectively, and “Total Variation” shows the percentage change in resistance at a 90° bend.

Three models were tested for assembling a resistive film with an elastomeric substrate:Model I: The resistive film was placed on top of the elastomeric substrate with the micro-pyramidal structure oriented tip up (see Figure 4a).Model II: The resistive film was placed above the elastomeric substrate with the micro-pyramidal structure oriented tip down (see Figure 4b).Model III: The resistive film was placed on an elastomeric substrate featuring an interlocked, paired pyramid structure (see Figure 4c).

At the outset of the study, we evaluated the Model I assembly. The meander resistor printed on ABS and PET exhibited a non-linear response (see Figure 5a,b), leading to its exclusion from subsequent tests. This may be due to the thermal degradation of the polymers. However, the response of the meander resistor printed on Kapton is depicted in Figure 5c.

Cyclic compression tests were performed, 20 cycles each, on the meander resistors printed on Kapton (model R1 v2); the assembly mode used was Model I, and the tests were performed at a compression force of 50 N/cm^2^ and a speed compression of 1 mm/min. An example of the answer to these tests can be seen in Figure 6.

Compression tests were performed on the printed meander resistor on Kapton (model R1 v3), assembled in all three assembly modes, to study the influence of the assembly mode on the piezoresistive response of the printed meander resistor. The test conditions were as follows: resistor model R1 v3, 50 N/cm^2^, 1 mm/min, 20 cycles, maximal load applied for 1 s. The results can be seen in Figure 7.

The piezoresistive response of the meander resistor model R1 v3, to compression, is shown in Figure 8, under the following conditions: 50 N/cm^2^, 1 mm/min, for all three assembly models—Model I, Model II, and Model III.

After this comparative testing of the three assembly models, it turns out that the assembly models Model II and Model III offer the best piezoresistive response. Further on, the tests will be performed only for these two assembly models.

We optimized our technology by substituting the silver paste used for soldering wires. The original paste required a curing temperature of 125 °C, whereas the new paste cures at just 70 °C. We then compared the piezoresistive response of the meander resistor model R1 before and after this optimization. The tests were conducted on resistor model R1 v3 under conditions of 50 N/cm^2^, at a speed of 1 mm/min, using assembly models II and III. The results are shown in Figure 9a,b.

In this study, we observed that optimization significantly improved the linearity of the piezoresistive response. Post-optimization, the Model III assembly exhibited superior piezoresistive characteristics. We also examined the piezoresistive response of a meander resistor printed on a flexible polyimide substrate. The tests revealed a completely non-linear response, leading to its exclusion from further studies.

The final phase of the research focused on the impact of the resistivity of screen printing inks on the piezoresistive response of meander resistors printed on Kapton R1 v3. We conducted comparative tests using two different inks—Ink1 and Ink2—under identical conditions, as follows: resistors printed on Kapton R1 v3, subjected to 50 N/cm^2^ pressure, and a 1 mm/min test speed, using both Model II and Model III assembly models, as shown in Figure 10. The results indicated that the sensors printed with Ink 1 and Ink 2 on Kapton exhibited nearly identical piezoresistive responses.

Figure 11 show the piezoresistive response of the resistor printed on Kapton R1 v3, with Ink 2, to cyclic tests of 20 cycles, 50 N/cm^2^, 5 mm/min, for the two assembly models Model II and Model III, respectively.

The optimization process enhanced sensor performance by improving linearity. While the polyimide substrate was unsuitable due to non-linear responses, the Kapton R1 v3 substrate with both tested inks performed reliably, ensuring consistent piezoresistive responses under the specified test conditions.

## 4. Conclusions

In this study, the impact of various flexible substrates on the performance of a meander resistor and the effect of elastomeric substrates on a proposed flexible pressure sensor were explored. The findings of this study revealed that the meander resistor, when deposited on a flexible Kapton substrate, demonstrated superior linearity in response. Further investigations indicated that the optimal elastomeric substrate combination involved the Model II and Model III assembly configurations, both of which exhibited nearly identical responses during cyclic testing. Optimization efforts led to significant improvements in response under low compression pressures, specifically for tests conducted up to 50 N/cm^2^. The introduction of Ink 2 did not yield substantial enhancements; however, it resulted in a higher value of the printed meander resistors. In conclusion, the optimal configuration was determined to be the meander resistor on a Kapton substrate with the Model II assembly model. Both Ink 1 and Ink 2 were deemed suitable for use. For optimal performance, the terminal wires should be soldered with silver paste and cured at 70 °C to maintain the integrity of the Ink 1 ink. Future work will involve developing a series of these flexible pressure sensors and evaluating their performance in a Wheatstone bridge configuration to further enhance their applicability and performance metrics.

Sensor Performance

Linearity of Response: The Kapton substrate significantly improved the linearity of the meander resistor’s response, making it the preferred choice for the sensor’s base material.Elastomeric Substrate Optimization: Model II and Model III assembly models provided the best results, showing a consistent and reliable performance under cyclic loading.Pressure Response: The optimized sensor showed enhanced sensitivity to low compression pressures, up to 50 N/cm^2^, ensuring accurate pressure measurements in this range.Ink Performance: While Ink 2 increased the resistor’s value, it did not notably improve overall performance, indicating that both Ink 1 and Ink 2 are viable options for the sensor.Assembly and Soldering: The use of silver paste for soldering at 70 °C was crucial to preserving the behaviour and performance of Ink 1, highlighting the importance of the assembly process in sensor fabrication.

By focusing on these optimized configurations and materials, we aim to enhance the performance and reliability of flexible pressure sensors, paving the way for their application in various fields. The next stage of development will involve integrating these sensors into a Wheatstone bridge configuration to further test and refine their performance characteristics.

## Figures and Tables

**Figure 1 micromachines-15-00847-f001:**
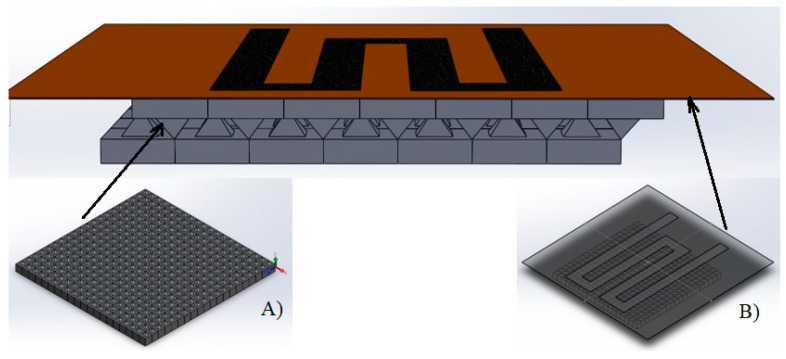
The architecture of the proposed pressure sensor and details. (**A**) The substrate and (**B**) the KAPTON Foil with the printed sensor.

**Figure 2 micromachines-15-00847-f002:**
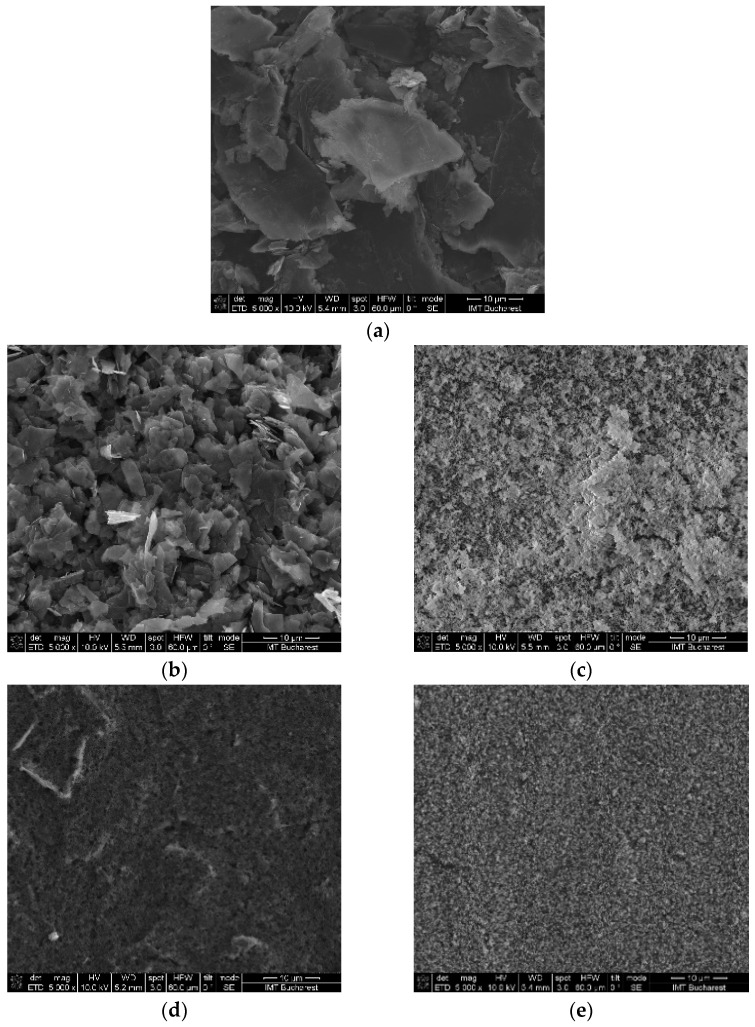
SEM images of nanomaterials used in the ink formulations, as well as the printed layers’ surfaces. (**a**) graphite; (**b**) Amorphous carbon; (**c**) TiO_2_; (**d**) Ink 1 on polyamide; (**e**) Ink 2 on Kapton substrate. Scale 10 µm.

**Figure 3 micromachines-15-00847-f003:**
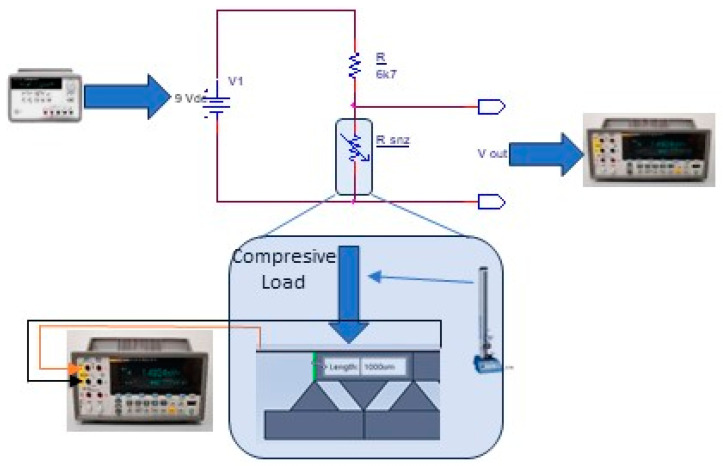
Diagram for tests of the resistive film assembly with elastomeric substrate, for compression.

**Figure 4 micromachines-15-00847-f004:**
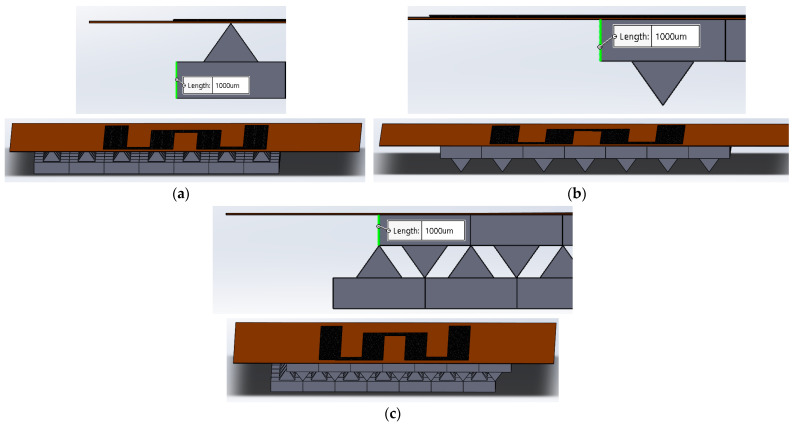
(**a**) Model I assembly. (**b**) Model II assembly. (**c**) Model III assembly.

**Figure 5 micromachines-15-00847-f005:**
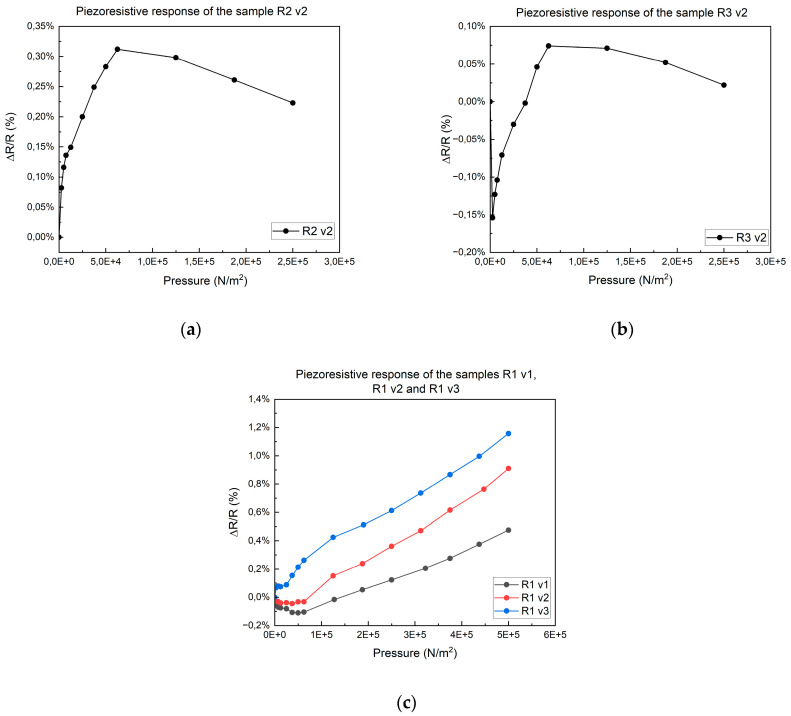
The non-linear response of the meander resistor. (**a**) Model R2 v2. (**b**) Model R3 v2. (**c**) Response given by three replicas of meander resistors printed on Kapton.

**Figure 6 micromachines-15-00847-f006:**
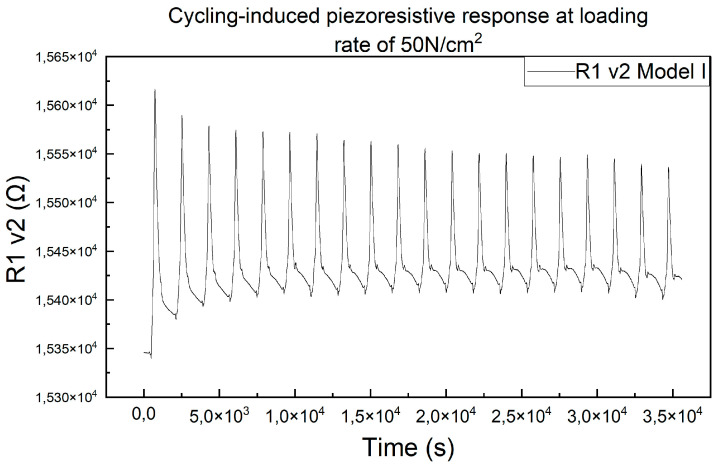
Piezoresistive response of R1 v2 tested in Model I assembly mode, 50 N/cm^2^ and 1 mm/min.

**Figure 7 micromachines-15-00847-f007:**
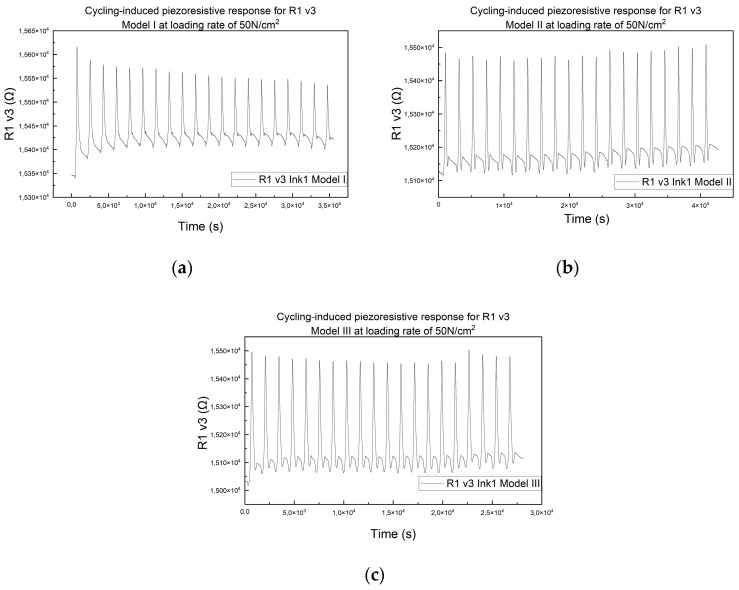
(**a**) Piezoresistive response of resistor R1 v3, assembly mode Model I, 50 N/cm^2^, 1 mm/min. (**b**) Piezoresistive response of resistor R1 v3, assembly mode Model II, 50 N/cm^2^, 1 mm/min. (**c**) Piezoresistive response of resistor R1 v3, assembly mode Model III, 50 N/cm^2^, 1 mm/min.

**Figure 8 micromachines-15-00847-f008:**
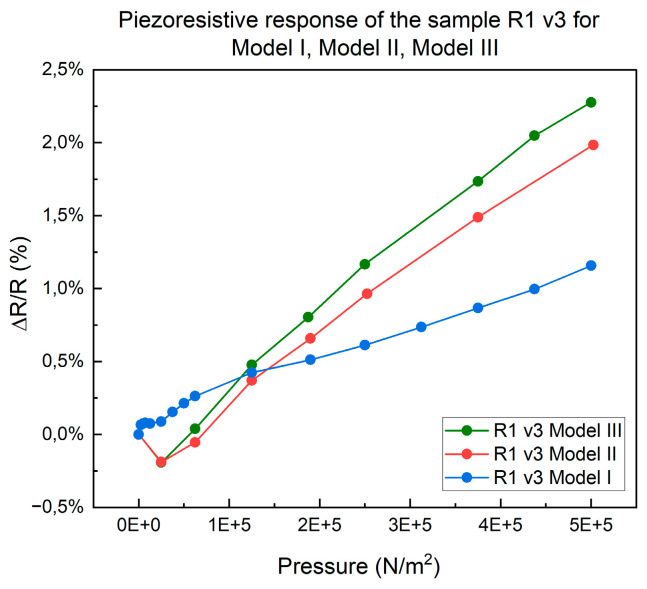
The response given by the resistor R1 v3, for the assembly modes—Model I, Model II, and Model III.

**Figure 9 micromachines-15-00847-f009:**
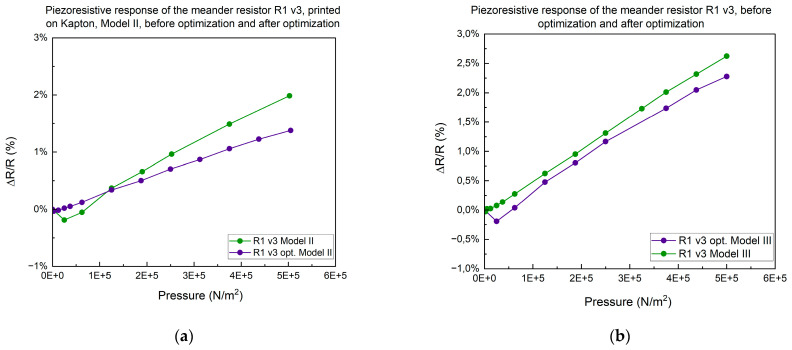
(**a**) Response of meander resistor printed on Kapton R1 v3, before optimization and after optimization, for the Model II assembly model. (**b**) Response of meander resistor printed on Kapton R1 v3, before optimization and after optimization, for the Model III assembly model.

**Figure 10 micromachines-15-00847-f010:**
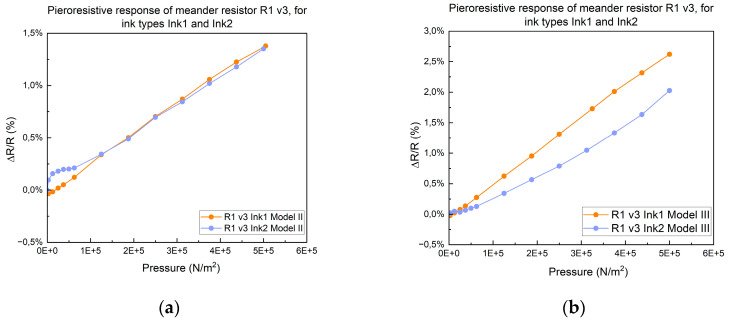
The piezoresistive response given by the meander resistor R1 v3, with Ink 1 and Ink 2, respectively. (**a**) Assembly Model II and (**b**) Model III.

**Figure 11 micromachines-15-00847-f011:**
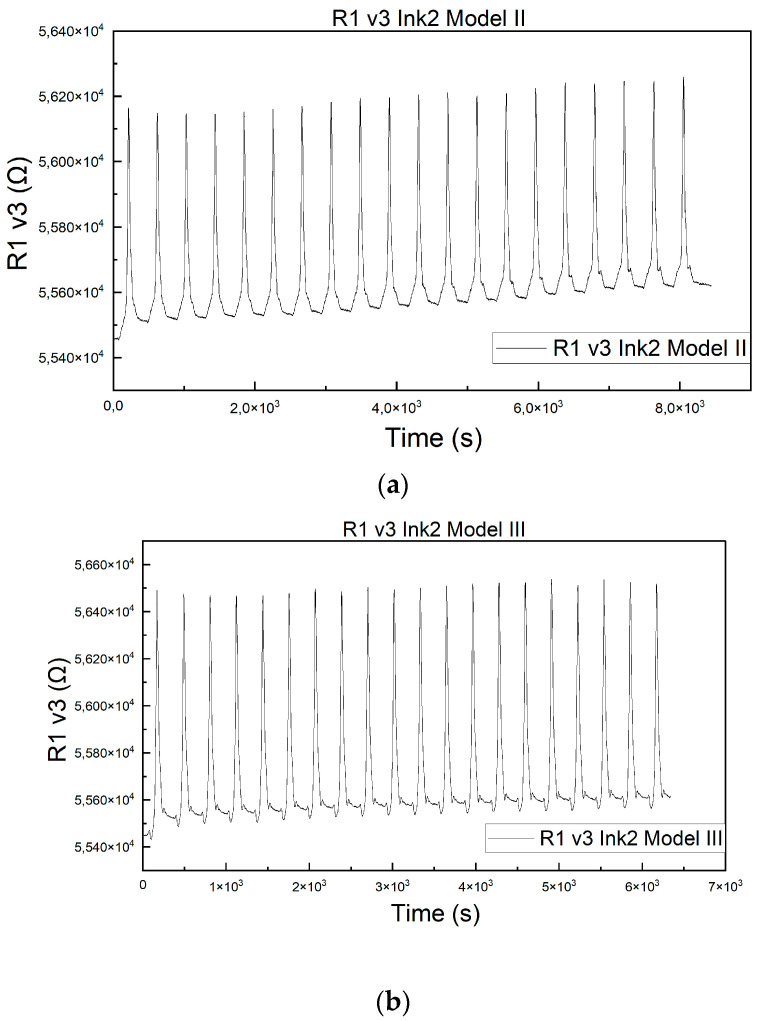
Piezoresistive response of R1 v3, Ink 2, (**a**) Model II, (**b**) Model III assembly model at 50 N/cm^2^, 20 cycles, 5 mm/min.

**Table 1 micromachines-15-00847-t001:** Resistor values before and after annealing.

Resistor Number	Resistor Version (Replication)	Initial Value Ω	Final Value Ω	Variation of the Resistor Value %
R1	V1	14,587.0 ± 1.45	17,804.0 ± 1.78	22.05
	V2	12,168.0 ± 1.21	14,733.0 ± 1.47	21.08
	V3	12,641.0 ± 1.26	15,040.0 ± 1.5	18.98
R3	V1	8117.0 ± 0.81	11,232.0 ± 1.12	38.38
	V2	6350.0 ± 0.63	8836.0 ± 0.88	39.15
R2	V1	7517.0 ± 0.75	9758.0 ± 0.97	29.81
	V2	2709.0 ± 0.27	4424.0 ± 0.44	63.31
	V3	6468.0 ± 0.64	8030.0 ± 0.8	24.14
R4	V1	5602.0 ± 0.56	-	-
	V2	9840.0 ± 0.98	-	-
	V3	7480.0 ± 0.74	-	-

**Table 2 micromachines-15-00847-t002:** Detailed data of resistance values at bending for resistors onto various substrates.

Material	Initial Value Ω	Bending 45° Ω	Bending 90° Ω	Total Variation %
Kapton	14,400.0 ± 1.44	14,250.0 ± 1.42	14,080.0 ± 1.40	2222
ABS	11,042.0 ± 1.10	10,827.0 ± 1.08	10,560.0 ± 1.05	4365
Kapton	14,808.0 ± 1.48	14,616.0 ± 1.46	14,485.0 ± 1.44	2181
PET	9615.0 ± 0.96	9384.0 ± 0.93	9180.0 ± 0.91	4524

## Data Availability

The raw and processed data required to reproduce these findings cannot be shared at this time due to technical or time limitations. The raw and processed data will be provided upon reasonable request to the corresponding author until the technical problems have been solved.

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
