# Peer review of "On the Development of a New Flexible Pressure Sensor"

_micromachines, 2024, doi:10.3390/mi15070847_

Round 1
Reviewer 1 Report
Comments and Suggestions for Authors
1 Figure 1 and Figure 2 exhibit the same structure. I think figure 1 is necessary.
2 Figure 3 is ugly. Figure 3 should be processed with better contrast or brightness settings. And the scale bar should be labeled by hand to clearly show the size. Figure 3(a) is graphite, but in the figure note, figure 3(b) is graphite. You did not use graphite in your experiment, so why you show the SEM image of graphite?
3 Figure 4 should show where is the wire connected on the sensor? The structure of the sensor is consists of two layers of PDMS with micro-pyramids, there is one layer of pyramid in this figure, why?
4 Normally, the resistant is declined after thermal annealing. Why the resistance is increased in Table 1?
5 Why the meander resistor printed on ABS and PET response different from that on Kapton substrate? Why there are 3 curves in Figure 6c? Please state clearly what are these curves stand for?
6 What is the meaning of R1, R2, v1, v2? Does the processing parameter is different? If the sample is fabricated using the same parameter and processes, you can not name it as different samples. It is just show the unreliable of your sensor.
7 Figure 6-12 are ugly, please redraw these figures.
8 In figure 7 and 8, the X-coordinate should be the number of cycle.
9 Why you give figure7a and 7b? What is the difference between sample R1v3 and R1v2? The curve is apparently different.
10 For Figure 9 and 10, the X-coordinate should be the pressure.
11 In figure 8c, there is a platform in every cycle, why the phenomenon appears?
13 Why the unit of loading rate is N/cm2?
14 There are lots of grammar or description mistakes.
For example, in line 135, SEM microscopy is wrong. The full name of SEM is scanning electron microscopy. It is not proper to use SEM microscopy.
Comments on the Quality of English Language
There are lots of grammar or description mistakes.
For example, in line 135, SEM microscopy is wrong. The full name of SEM is scanning electron microscopy. It is not proper to use SEM microscopy.
The manuscript should be rewrite.
Author Response
We thank the reviewer for their time and effort. The manuscript was revised in according to the comments. Most of the constructive suggestions and comments were considered and the manuscript was improved. Detailed answers are provided bellow. Due to some problems of incompatibility of the journal template with our MS Office version, the manuscript format has still some problems for which we would ask mdpi team further help on fixing it. We hope that this revised version is now ready to be accepted.
Answers to comments
1 Figure 1 and Figure 2 exhibit the same structure. I think figure 1 is necessary.
We have combined the two figures and improved the quality. Please see the improved version of the manuscript.
2 Figure 3 is ugly. Figure 3 should be processed with better contrast or brightness settings. And the scale bar should be labeled by hand to clearly show the size. Figure 3(a) is graphite, but in the figure note, figure 3(b) is graphite. You did not use graphite in your experiment, so why you show the SEM image of graphite?
We have improved the quality of figure 3, now existing as figure 2. At least in the word version the images are clear and of high resolution on our screen. The scale bar cannot be modified. That will be altering of the original data. We added the information regarding the scale size in the figure caption. Graphite is a component of the carbon micro-powder (graphite and C black). The text was corrected. Please see the improved version of the manuscript.
3 Figure 4 should show where is the wire connected on the sensor? The structure of the sensor is consists of two layers of PDMS with micro-pyramids, there is one layer of pyramid in this figure, why?
We have improved the quality of the figure 4 (now 3) and added the place of wire connection. Please see the improved version of the manuscript.
4 Normally, the resistant is declined after thermal annealing. Why the resistance is increased in Table 1?
We were also surprised by these results, however, since the behaviour was consistent on all samples, we presented the real measurements results. This may be a consequence of solvent evaporation. We added this hypothesis in the text. Please see the improved version of the manuscript.
5 Why the meander resistor printed on ABS and PET response different from that on Kapton substrate? Why there are 3 curves in Figure 6c? Please state clearly what are these curves stand for?
The meander resistor printed on ABS and PET response is different from that on Kapton substrate since the ABS and PET cannot survive the thermal treatment while the Kapton does. The explanation was added to the manuscript. Also, the explanation for the 3 curves in fig 6c (now 5) was added. Please see the improved version of the manuscript.
6 What is the meaning of R1, R2, v1, v2? Does the processing parameter is different? If the sample is fabricated using the same parameter and processes, you can not name it as different samples. It is just show the unreliable of your sensor.
To verify the reliability and repeatability of the production process, 3 samples from each type were prepared and they are called V (Versions of the same sample). The differences on their resistance are of Ω magnitude, actually a very good achievement for the specific materials and fabrication method.
7 Figure 6-12 are ugly, please redraw these figures.
The figures were improved. Now they are fig 5-11. Please see the improved version of the manuscript.
8 In figure 7 and 8, the X-coordinate should be the number of cycle.
In the x coordinate is the time. This is the way that the test equipment ( MECMESIN) provide the results and changes would represent data manipulation. We prefer to keep the X scale according to the original data. We hope that the reviewer agrees.
9 Why you give figure7a and 7b? What is the difference between sample R1v3 and R1v2? The curve is apparently different.
We thank to the reviewer for the observation. The problem was fixed in the revised version of the manuscript.
10 For Figure 9 and 10, the X-coordinate should be the pressure.
Thank you for the observation. It was corrected in the revised version of the manuscript.
11 In figure 8c, there is a platform in every cycle, why the phenomenon appears?
It was a consequence of the relative values that were used. The revised manuscript present R variations instead of the relative ones. Please see the improved version of the manuscript.
13 Why the unit of loading rate is N/cm2?
The loading rate was changed to SI units in the revised manuscript.
14 There are lots of grammar or description mistakes.
For example, in line 135, SEM microscopy is wrong. The full name of SEM is scanning electron microscopy. It is not proper to use SEM microscopy.
Comments on the Quality of English Language
There are lots of grammar or description mistakes.
For example, in line 135, SEM microscopy is wrong. The full name of SEM is scanning electron microscopy. It is not proper to use SEM microscopy.
The manuscript should be rewrite.
The manuscript was corrected. We believe that now is OK. Anyway, if the case, small English corrections will be performed also during proofing with the MDPI team after the manuscript is accepted. We hope that the reviewer agrees on this.
Reviewer 2 Report
Comments and Suggestions for Authors
In the graphs when refered to the applied pressure (N/cm2) is is write power instead of pressure.
The graphs quality is relatively poor. Graphs sizes are not uniform.
The process of applying the cyclic pressure is not well described: the pressure of 50 N/cm2 seems to be the final pressure but it not clear wheter it is returned to zero in each cycle. Besides, it is described that it is achieved by deflecting at a rate of 1 mm/min: It is not clear the relationship between the deflection and the pressure applied. The total thickness of the devices are not specified clearly, but seems to be around 1,5 millimeters, then , how is actually the variation of pressure done. this shoud be explained clearly. If instead the mm in the 1mm/min, means millimetre of mercury, the units of pressure along the paper should be uniformized.
It should be commented on how long the maximum stress is applied and if this results in plastic deformations of the components of the device, particularly in the sensitive layer (It is assumed that the substrate does not suffer this.
In some graphs, negative variations in the change in resistance are observed. What is this due to?
The observed variations are small (~2%), how is this taken into account with the tolerances and possible variations due to temperature? Do not forget that it is basically a carbon film resistor, which has high temperature coefficients.
Author Response
We thank the reviewer for their time and effort. The manuscript was revised in according to the comments. Most of the constructive suggestions and comments were considered and the manuscript was improved. Detailed answers are provided bellow. Due to some problems of incompatibility of the journal template with our MS Office version, the manuscript format has still some problems for which we would ask mdpi team further help on fixing it. We hope that this revised version is now ready to be accepted.
Answers to comments
In the graphs when refered to the applied pressure (N/cm2) is is write power instead of pressure.
Thank you for the observation. It was corrected in the revised manuscript.
The graphs quality is relatively poor. Graphs sizes are not uniform.
Thank you for the observation. All figures and the graphs were improved in the revised manuscript.
The process of applying the cyclic pressure is not well described: the pressure of 50 N/cm2 seems to be the final pressure but it not clear wheter it is returned to zero in each cycle. Besides, it is described that it is achieved by deflecting at a rate of 1 mm/min: It is not clear the relationship between the deflection and the pressure applied. The total thickness of the devices are not specified clearly, but seems to be around 1,5 millimeters, then , how is actually the variation of pressure done. this shoud be explained clearly. If instead the mm in the 1mm/min, means millimetre of mercury, the units of pressure along the paper should be uniformized.
The Mecmesin ( testing machine) is returning to 0 in each cycle, however in the graph was represented ∆R/R not R. The quality of the graphs was improved in the revised manuscript. W hope that now is clearer.
It should be commented on how long the maximum stress is applied and if this results in plastic deformations of the components of the device, particularly in the sensitive layer (It is assumed that the substrate does not suffer this.
The maximum stress is applied for 1 second time. Supplementary information was added in the revised manuscript text. We hope that now is OK.
In some graphs, negative variations in the change in resistance are observed. What is this due to?
It was because of the use of ∆R/R not R. The revised manuscript shows R variations instead of the relative values.
The observed variations are small (~2%), how is this taken into account with the tolerances and possible variations due to temperature? Do not forget that it is basically a carbon film resistor, which has high temperature coefficients.
The reviewer is right but it was used ∆R/R not R. The revised manuscript was improved and shows R variations instead of the relative values.